# Mental-somatic multimorbidity in trajectories of cognitive function for middle-aged and older adults

Siting Chen[1], Corey L. Nagel[2], Ruotong Liu[3], Anda Botoseneanu[4,5], Heather G. Allore[6,7], Jason T. Newsom[8], Stephen Thielke[9], Jeffrey Kaye[10], Ana R. Quiñones[1,3]*

1 OHSU-PSU School of Public Health, Portland, Oregon, United States of America, 2 College of Nursing, University of Arkansas for Medical Sciences, Little Rock, Arkansas, United States of America, 3 Department of Family Medicine, Oregon Health & Science University, Portland, Oregon, United States of America, 4 Department of Health & Human Services, University of Michigan, Dearborn, Michigan, United States of America, 5 Institute of Gerontology, University of Michigan, Ann Arbor, Michigan, United States of America, 6 Department of Internal Medicine, Yale University, New Haven, Connecticut, United States of America, 7 Department of Biostatistics, Yale University, New Haven, Connecticut, United States of America, 8 Department of Psychology, Portland State University, Portland, Oregon, United States of America, 9 Department of Psychiatry and Behavioral Sciences, University of Washington, Seattle, Washington, United States of America, 10 Department of Neurology, Oregon Health & Science University, Portland, Oregon, United States of America

* quinones@ohsu.edu

**Data Availability Statement:** The data underlying the results presented in this study are publicly available from the Health and Retirement Study, http://hrsonline.isr.umich.edu/. All analytic data

## Abstract

### Introduction

Multimorbidity may confer higher risk for cognitive decline than any single constituent disease. This study aims to identify distinct trajectories of cognitive impairment probability among middle-aged and older adults, and to assess the effect of changes in mental-somatic multimorbidity on these distinct trajectories.

### Methods

Data from the Health and Retirement Study (1998–2016) were employed to estimate group-based trajectory models identifying distinct trajectories of cognitive impairment probability. Four time-varying mental-somatic multimorbidity combinations (somatic, stroke, depressive, stroke and depressive) were examined for their association with observed trajectories of cognitive impairment probability with age. Multinomial logistic regression analysis was conducted to quantify the association of sociodemographic and health-related factors with trajectory group membership.

### Results

Respondents (N = 20,070) had a mean age of 61.0 years (SD = 8.7) at baseline. Three distinct cognitive trajectories were identified using group-based trajectory modelling: (1) **Low risk with late-life increase** (62.6%), (2) **Low initial risk with rapid increase** (25.7%), and (3) **High risk** (11.7%). For adults following along **Low risk with late-life increase**, the odds of cognitive impairment for stroke and depressive multimorbidity (OR:3.92, 95%

files are available from the Figshare database (accession number(s) https://doi.org/10.6084/m9.figshare.25631778.v1).

**Funding:** This work was supported by the National Institute on Aging at the National Institutes of Health (grant numbers RF1AG058545 to ARQ; HGA who contributed from the Yale Claude D. Pepper Older Americans Independence Center P30AG021342 and Yale Alzheimer's Disease Research Center P30AG066508; P30AG066518, and P30AG024978 to JK). Content is solely the responsibility of the authors and does not necessarily represent official views of the National Institutes of Health. The funders played no role in the design, execution, analysis, or interpretation of the data or writing of the study.

**Competing interests:** The authors have declared that no competing interests exist.

CI:2.91,5.28) were nearly two times higher than either stroke multimorbidity (OR:2.06, 95% CI:1.75,2.43) or depressive multimorbidity (OR:2.03, 95%CI:1.71,2.41). The odds of cognitive impairment for stroke and depressive multimorbidity in **Low initial risk with rapid increase** or **High risk** (OR:4.31, 95%CI:3.50,5.31; OR:3.43, 95%CI:2.07,5.66, respectively) were moderately higher than stroke multimorbidity (OR:2.71, 95%CI:2.35, 3.13; OR: 3.23, 95%CI:2.16, 4.81, respectively). In the multinomial logistic regression model, non-Hispanic Black and Hispanic respondents had higher odds of being in **Low initial risk with rapid increase** and **High risk** relative to non-Hispanic White adults.

## Conclusions

These findings show that depressive and stroke multimorbidity combinations have the greatest association with rapid cognitive declines and their prevention may postpone these declines, especially in socially disadvantaged and minoritized groups.

## 1. Introduction

Cognitive decline is a prominent feature within the continuum of Alzheimer's disease and related dementias (ADRD), and poses challenging and complex problems exerting considerable health, social, and psychological burdens on individuals, and high costs to societies [1, 2]. The efficacy of early interventions intended to delay cognitive decline resulting from the progression of ADRD is supported by existing evidence, although approved disease-modifying treatments are scarce [1]. Therefore, it is critical to identify potentially modifiable risk factors and to inform the development of feasible interventions for older adults that can be implemented during critical, transitional stages of cognitive decline [3–5].

Multimorbidity (≥2 chronic diseases) commonly occurs among older adults [6] and is associated with a higher risk of cognitive decline and ADRD [6–8]. However, many studies conceptualize multimorbidity as a count of chronic diseases or as a single summary index/score, making it difficult to assess the impact of specific combinations of diseases [9–11]. In addition, prior studies have largely been conducted using cross-sectional designs or with short follow-up periods, which precludes the identification of long-term associations between multimorbidity and cognitive trajectories [9–11].

More recent work examines specific multimorbidity combinations [8, 12–16] to address questions regarding chronic disease contributions to adverse outcomes, including cognitive decline. The U.S. Department of Health and Human Services (HHS) outlined a conceptual framework for considering both somatic and mental health conditions in defining and measuring multimorbidity [17]. In particular, stroke represents one of the leading causes of cognitive impairment among adults with cardiometabolic conditions [18, 19], while depression, one of the most prevalent mental health disorders, may confer an increased risk of dementia [20, 21]. Several studies address the association between cardiometabolic multimorbidity—including stroke—as a critical component and cognitive decline [12, 14], and investigate the specific effect of depression on cognitive impairment in the context of co-existing morbidities [22, 23]. While these findings have shed light on possible shared mechanisms and pathways between multiple, co-occurring diseases that may contribute to the development of cognitive impairment, there is a paucity of research examining the combinations of diseases on cognitive impairment in comparison with either condition individually, or in the absence of both.

Therefore, it is of considerable interest to examine mental-somatic multimorbidity profiles more broadly to elucidate differential associations with cognitive impairment.

The aim of this study is threefold. First, we identify and characterize distinct trajectories of the probability of cognitive impairment with advancing age among a large, nationally-representative cohort of middle-aged and older Americans. Second, we assess the differential association of changing morbidity profiles among four mental-somatic multimorbidity combination categories on the probability of cognitive impairment for each identified trajectory group. Third, we examine the sociodemographic and health-related characteristics associated with the probabilities of membership to each of the identified trajectory groups.

## 2. Materials and methods

### 2.1. Data source

The Health and Retirement Study (HRS) is an ongoing, nationally-representative longitudinal survey of noninstitutionalized Americans ages 51 and older. Interviews are conducted biennially to evaluate the health and economic standing of respondents toward the end of their work life and into retirement [24]. We used HRS survey waves 1998–2016 in this study to ensure measurement concordance. These data have been previously collected and are publicly available and, therefore, fully anonymized. The study protocol was approved by the Oregon Health and Science University Institutional Review Board under exemption category 4 (without need to obtain prior consent). The data were assessed from March 2022 to July 2023 for this current study.

### 2.2. Study population

The current study followed HRS participants from the earliest age of cohort eligibility until dropout or death. Of the 35,689 HRS respondents interviewed between 1998 and 2016 who were living in the community and cohort-eligible (i.e., respondents with a positive survey weight), we excluded 1,257 participants with proxy respondents due to missing assessment of depressive symptoms and 5,357 participants who reported other race or had missing data on any covariate or inconsistent reporting on chronic diseases after adjudication (i.e., "yes" followed by "no" at subsequent waves) [25]. Lastly, an additional 9,005 participants with fewer than 3 assessments of cognitive function and/or chronic diseases, as required for adequate modeling of temporal trajectories, were also excluded. The final analytic sample consisted of 20,070 respondents. The details of the study sample flow diagram are shown and described in **S1 Fig** in S1 Appendix.

### 2.3. Measures

**2.3.1 Primary outcome: Cognitive impairment.**   Cognitive function was measured at each wave using the 27-point HRS cognitive scale [26, 27], a modified version of the Telephone Interview for Cognitive Status (TICS) [28]. This assessment includes 1) an immediate and delayed free recall test (range 0–20); 2) a serial sevens subtraction test (range 0–5); and 3) a counting backwards test (range 0–2). The summary cognitive function score across the composite subscales ranges between 0–27, with higher scores indicating better cognitive function. Consistent with Langa-Weir classification [29], the continuous score was categorized into three derived categories of cognitive function: normal (range 12–27), cognitively impaired but not demented (CIND) (range 7–11), and demented (range 0–6). For this analysis, we collapsed the CIND and demented categories to construct a binary indicator of *normal* versus *impaired* cognition.

**2.3.2 Independent time-varying covariate:Mental-somatic multimorbidity combinations.** Information on seven self-reported, physician-diagnosed chronic somatic conditions was collected at each interview: heart disease, hypertension, stroke (but not transient ischemic attack), diabetes, arthritis, lung disease, and cancer. These were assessed at baseline with, "Has a doctor ever told you that you have. . .?", and at follow-up waves with, "Since we last talked with you, has a doctor told you that you have. . .?". Depressive symptoms were measured at each wave using the 8-item Centers for Epidemiologic Research Depression (Center for Epidemiological Studies-Depression scale [CES-D 8]) scale [30, 31]. Respondents with four or more symptoms were defined as having high depressive symptoms [32].

Chronic disease multimorbidity was modeled as a time-varying variable, categorized at each wave as no multimorbidity (no or only one disease) or one of four mutually-exclusive multimorbidity combinations: 1) somatic multimorbidity excluding stroke ($\geq$2 diseases: heart disease, lung disease, hypertension, arthritis, diabetes, cancer); 2) stroke multimorbidity (stroke and $\geq$1 somatic disease); 3) depressive multimorbidity (high depressive symptoms and $\geq$1 somatic disease excluding stroke); 4) stroke and depressive multimorbidity (both stroke and high depressive symptoms with/without any other somatic disease that may be present). These multimorbidity combinations were included in the model as time-varying covariates, such that participants could accumulate conditions and advance to a higher multimorbidity category (e.g., from stroke multimorbidity to stroke and depressive multimorbidity) or could revert to a lower multimorbidity category due to lower depressive symptoms in the subsequent waves.

**2.3.3 Covariates.** The following sociodemographic and health-related covariates were measured at the baseline interview: race/ethnicity (mutually exclusive categories: non-Hispanic White, non-Hispanic Black, Hispanic); sex (female/male); highest education (<high school, high school graduate, some college or $\geq$college graduate); household wealth (quartiles derived from baseline net worth in US dollars); smoking status (current, past, never smoker), and body mass index (BMI) category (underweight, healthy weight, overweight, obese). Specifically, race/ethnicity was defined using the two following questions: 1) "Do you consider yourself Hispanic or Latino?" and 2) "Do you consider yourself primarily white or Caucasian, Black or African American, American Indian, or Asian, or something else?" If the respondent identified as Hispanic, this would be prioritized over any other racial categories and the respondent would be categorized as Hispanic. Three mutually-exclusive groups were constructed for the analyses: non-Hispanic white, non-Hispanic black, and Hispanic. The BMI categories were defined as underweight (BMI<18.5), healthy weight (BMI = 18.5 to <25.0), overweight (BMI = 25 to <30.0), and obese (BMI $\geq$30) [33].

## 2.4. Statistical analysis

**2.4.1 Trajectories of cognitive impairment and model selection.** Group-based trajectory modeling (GBTM) is a semi-parametric, finite mixture modeling approach that uses maximum likelihood estimation to identify groups of individuals following trajectories of a similar pattern [34]. Centered age was the time metric for our analysis. We selected a logit link for GBTMs given the binary outcome, resulting in trajectories that represent the predicted probability of cognitive impairment with advancing age for each identified group. Following established guidance [35], we began by fitting a sequence of unconditional GBTMs in order to 1) determine the optimal number of trajectory groups and 2) select the most appropriate functional form (intercept only, linear, quadratic, or cubic) of each trajectory group. Model selection was an iterative process based on a combination of following criteria[35]: 1) diagnostic assessments including reduction in Bayesian Information Criterion (BIC), average posterior

probability of group membership > 80% for all groups, odds of correct classification >5.0; 2) size of the smallest group >10% of total sample; and 3) the ability to capture clinically relevant and distinct trajectories of cognitive impairment risk across the entire observed age span. Based on these criteria, we opted for the three-group model as the best solution. Detailed modeling processes, diagnostic statistics and trajectory plots are shown in **S2 Table in** S1 Appendix and **S2 Fig** in S1 Appendix.

**2.4.2 Estimated trajectories of cognitive impairment probability with transition between multimorbidity combination groups.** After selecting the three-group model, we included time-varying indicators for multimorbidity combination groups to examine their association with the observed trajectory within each trajectory group while adjusting for baseline age. Additionally, to minimize bias from loss to follow-up, we adjusted for nonrandom participant attrition after three survey waves and conducted sensitivity analyses between models with and without accounting for missing data due to attrition [36] (details provided in **S3-S5 Tables** in S1 Appendix, **S3 Fig** in S1 Appendix). This full model provided group-specific estimates of whether time-varying multimorbidity combinations were associated with the course of the probability of cognitive impairment with advancing age.

Moreover, changes in multimorbidity combinations (e.g., transitioning from somatic multimorbidity to stroke multimorbidity) may have differential associations with the cognitive impairment probability across the age span. To examine these transitions, we fit models simulating transition between multimorbidity combinations at pre-specified ages. Specifically, we compared the predicted trajectories of cognitive impairment probability between respondents who transitioned from somatic multimorbidity to stroke multimorbidity, depressive multimorbidity, or stroke and depressive multimorbidity at decades of age (60,70,80 years) vs. those with consistent somatic multimorbidity (i.e., combinations that did not involve stroke or depressive symptoms) with advancing age.

**2.4.3 Multinomial regression models: Sociodemographic and health-related covariates of trajectory group membership.** Based on the full GBTM accounting for time-varying multimorbidity combinations and attrition, we assigned each respondent to a trajectory group for which they had the maximum posterior probability of membership. Descriptive methods were used to summarize characteristics by trajectory group membership: frequencies and percentages were calculated for categorical variables while means and standard deviations were calculated for continuous variables. Additionally, we performed chi-square tests and ANOVA tests to compare the characteristics between trajectory groups. We then conducted a separate multinomial logistic regression analysis to assess the association of sociodemographic and health-related covariates with trajectory group membership. The full multinomial logistic regression model was adjusted for baseline age, race/ethnicity, sex, education, wealth, smoking, and BMI categories. Two-way and three-way interaction terms between covariates were tested in the models. We constructed an additional multinomial logistic regression model with a person's posterior probability of group membership as weights in a sensitivity analysis (See **S6 Table** in S1 Appendix).

All statistical analyses were performed in STATA/SE 16.1 and GBTMs were fit using the 'traj' package [37]. Data visualizations of trajectories were performed in R 3.6.2. A statistically significant level was set at $p < 0.05$. Full and complete details of our methodological procedures are provided in the S1 Appendix. Technical details of the statistical procedures and codes for visualizing cognitive impairment trajectories are included in S2 Appendix.

## 3. Results

### 3.1. Sample characteristics

The analytic sample consisted of 20,070 respondents with a mean age of 61.0 years (SD = 8.7) at the baseline interview (Table 1). 57.5% of the study sample were female and 42.5% were male. Most respondents were non-Hispanic White (70.0%). Table 1 provides detailed descriptive information on the analytic sample at baseline. We provided an additional table presenting the distribution of trajectory groups by covariates in **S1 Table** in S1 Appendix.

### 3.2. Cognitive impairment trajectories

Fig 1 displays the three distinct trajectories of cognitive impairment probability in the unconditional GBTM without inclusion of time-varying multimorbidity indicators: **Low risk with late-life increase**, **Low initial risk with rapid increase**, and **High risk**. Specifically, the **Low risk with late-life increase** trajectory (63.7%), which represented the largest proportion of the study sample, displayed a low probability of cognitive impairment at ages 51–70 and exhibited a slowly increasing probability after age 70. However, **Low initial risk with rapid increase** (24.5%) started with a minimal probability of cognitive impairment at baseline age but experienced a rapid increase in the probability of impairment after age 60. Unlike the other two trajectories, **High risk** (11.8%) started, on average, with a high baseline probability of cognitive impairment and showed a steady increase throughout later ages.

### 3.3. Time-varying multimorbidity combinations and impact on developmental trajectories of cognitive impairment probability

Table 2 presents the group-specific estimates of the association between multimorbidity combinations and observed trajectories of cognitive impairment probability in the full model. In **Low risk with late-life increase**, stroke multimorbidity (OR: 2.06; 95%CI: 1.75, 2.43) and depressive multimorbidity (OR: 2.03; 95%CI: 1.71, 2.41) had similar higher odds of cognitive impairment relative to somatic multimorbidity. However, the odds of cognitive impairment for stroke and depressive multimorbidity (OR: 3.92; 95%CI: 2.91, 5.28) were nearly two times higher than either stroke multimorbidity or depressive multimorbidity. In **Low initial risk with rapid increase** and **High risk**, the odds of cognitive impairment were highest for stroke and depressive multimorbidity relative to somatic multimorbidity, although the odds for stroke multimorbidity (OR: 3.23; 95%CI: 2.16, 4.81) are similar to stroke and depressive multimorbidity (OR: 3.43; 95%CI: 2.07, 5.66) in the **High risk** trajectory.

### 3.4. Predicted trajectories of cognitive impairment probability with multimorbidity transition at pre-specified ages

GBTMs were constructed to estimate discontinuous changes in cognitive impairment probability associated with multimorbidity transitions at pre-specified ages. Fig 2 shows the predicted trajectories of cognitive impairment probability for respondents who transitioned from somatic multimorbidity to stroke/depressive/stroke and depressive multimorbidity (dashed lines) at decades of age (60, 70, 80 years) and respondents with consistent somatic multimorbidity with advancing age (reference group, solid lines). In **Low risk with late-life increase**, respondents who developed stroke and depressive multimorbidity in later life (age 80) experienced a moderate increase in the probability of impairment relative to somatic multimorbidity. Unlike the course observed in **Low risk with late-life increase**, relative to respondents with somatic multimorbidity, respondents developing stroke and depressive multimorbidity at ages 60, 70 and 80 exhibited significant increases in the probability of cognitive impairment in

**Table 1. General characteristics of study population at baseline interview, Health and Retirement Study (1998–2016).**

| | Total | Low risk with late-life increase | Low initial risk with rapid increase | High risk | p value |
|---|---|---|---|---|---|
| **N (%)** | 20070 | 12560 (62.6) | 5155 (25.7) | 2355 (11.7) | |
| **Baseline age (mean (SD))** | 61.0 (8.7) | 60.3 (8.5) | 62.9 (9.1) | 60.7 (7.9) | <0.01 |
| **Sex, n (%)** | | | | | <0.01 |
| Male | 8522 (42.5) | 5222 (41.6) | 2221 (43.1) | 1079 (45.8) | |
| Female | 11548 (57.5) | 7338 (58.4) | 2934 (56.9) | 1276 (54.2) | |
| **Race/ethnicity, n (%)** | | | | | <0.01 |
| Hispanic | 2320 (11.6) | 1032 (8.2) | 710 (13.8) | 578 (24.5) | |
| NH White | 14046 (70.0) | 10042 (80.0) | 3234 (62.7) | 770 (32.7) | |
| NH Black | 3704 (18.5) | 1486 (11.8) | 1211 (23.5) | 1007 (42.8) | |
| **Education, n (%)** | | | | | <0.01 |
| < High school | 4166 (20.8) | 1242 (9.9) | 1533 (29.7) | 1391 (59.1) | |
| High school graduate | 10566 (52.6) | 6863 (54.6) | 2864 (55.6) | 839 (35.6) | |
| College | 5338 (26.6) | 4455 (35.5) | 758 (14.7) | 125 (5.3) | |
| **Wealth quartiles[a], n (%)** | | | | | <0.01 |
| 1st quartile (lowest) | 5462 (27.2) | 2472 (19.7) | 1643 (31.9) | 1347 (57.2) | |
| 2nd quartile | 4922 (24.5) | 2861 (22.8) | 1515 (29.4) | 546 (23.2) | |
| 3rd quartile | 4877 (24.3) | 3416 (27.2) | 1143 (22.2) | 318 (13.5) | |
| 4th quartile (highest) | 4809 (24.0) | 3811 (30.3) | 854 (16.6) | 144 (6.1) | |
| **Smoking, n (%)** | | | | | <0.01 |
| Never smoker | 8468 (42.2) | 5511 (43.9) | 2058 (39.9) | 899 (38.2) | |
| Past smoker | 7797 (38.8) | 4976 (39.6) | 1993 (38.7) | 828 (35.2) | |
| Current smoker | 3805 (19.0) | 2073 (16.5) | 1104 (21.4) | 628 (26.7) | |
| **BMI categories, n (%)** | | | | | <0.01 |
| Underweight | 204 (1.0) | 125 (1.0) | 53 (1.0) | 26 (1.1) | |
| Normal weight | 5854 (29.2) | 3770 (30.0) | 1462 (28.4) | 622 (26.4) | |
| Overweight | 7874 (39.2) | 4995 (39.8) | 1998 (38.8) | 881 (37.4) | |
| Obese | 6138 (30.6) | 3670 (29.2) | 1642 (31.9) | 826 (35.1) | |
| **No. of chronic conditions, mean (SD)** | 1.5 (1.3) | 1.4 (1.2) | 1.8 (1.4) | 1.9 (1.5) | <0.01 |
| **Multimorbidity, n (%)** | | | | | <0.01 |
| No multimorbidity | 11134 (55.5) | 7561 (60.2) | 2471 (47.9) | 1102 (46.8) | |
| Somatic multimorbidity | 6245 (31.1) | 3784 (30.1) | 1756 (34.1) | 705 (29.9) | |
| Stroke multimorbidity | 537 (2.7) | 291 (2.3) | 178 (3.5) | 68 (2.9) | |
| Depressive multimorbidity | 1959 (9.8) | 860 (6.8) | 658 (12.8) | 441 (18.7) | |
| Stroke and Depressive multimorbidity | 195 (1.0) | 64 (0.5) | 92 (1.8) | 39 (1.7) | |
| **Cognitive Score, mean (SD)** | 16.1 (4.3) | 17.9 (3.1) | 14.3 (3.8) | 9.9 (3.5) | <0.01 |
| **Cognition category, n (%)** | | | | | <0.01 |
| Normal | 17057 (85.0) | 12446 (99.1) | 3995 (77.5) | 616 (26.2) | |
| Cognitive impairment (CIND or demented) | 3013 (15.0) | 114 (0.9) | 1160 (22.5) | 1739 (73.8) | |
| **Attrition, n (%)** | | | | | <0.01 |
| No attrition | 12198 (60.8) | 8120 (64.6) | 2752 (53.4) | 1326 (56.3) | |
| Attrition | 7872 (39.2) | 4440 (35.4) | 2403 (46.6) | 1029 (43.7) | |

Abbreviations: NH = non-Hispanic; BMI: body mass index; CIND: cognitively impaired but not demented.

[a]Quartiles for wealth were derived from baseline net worth in US dollars.

Note: Group membership is assigned for each participant based on the maximum posterior probability from the full group-based trajectory model (with time-varying multimorbidity, adjusted or baseline age, and accounting for attrition). Chi-square tests were performed for categorical variables. ANOVA tests were performed for continuous variables.

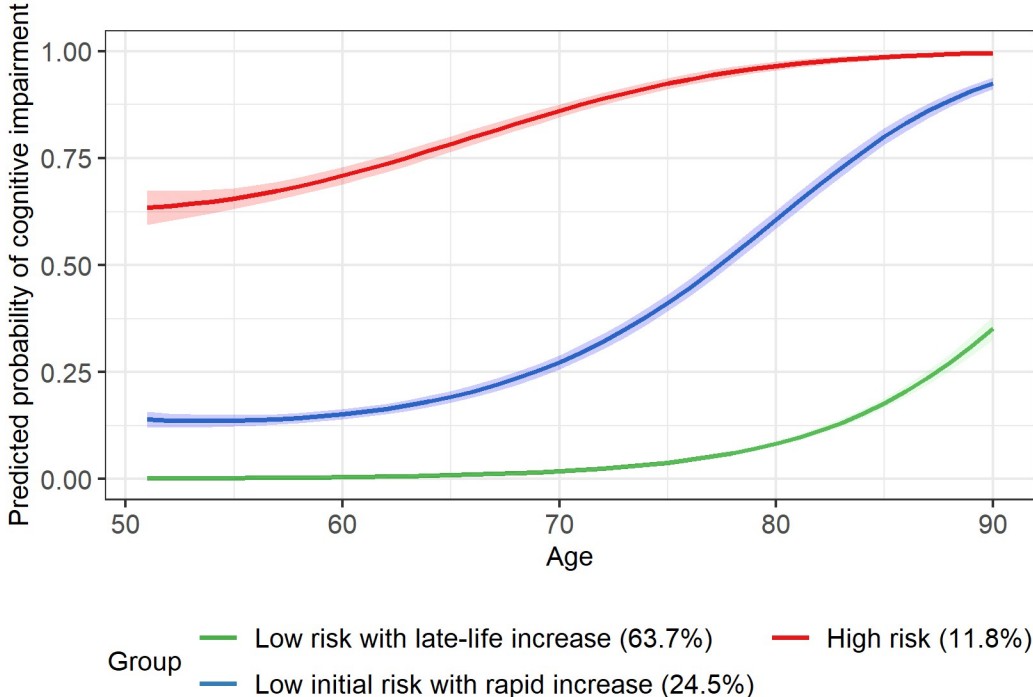

**Fig 1. Predicted probability of cognitive impairment in the unconditional group-based trajectory model.** Group membership is assigned for each participant based on maximum posterior probability rule from the unconditional group-based trajectory model (without time-varying multimorbidity, not accounting for attrition). Confidence bands represent 95% CIs of predicted probabilities.

the **Low initial risk with rapid increase** and **High risk** trajectories. Specifically, **Low initial risk with rapid increase** showed a large increase in probability of cognitive impairment when transitioning from somatic multimorbidity to stroke/depressive/stroke and depressive multi-morbidity. Group-specific estimates of cognitive impairment probabilities by multimorbidity groups were provided in **S7 Table** in S1 Appendix.

### 3.5. Multinomial regression models: socio-demographic and health-related covariates associated with trajectory group membership

In the multinomial logistic regression model with covariates (Table 3), non-Hispanic Black and Hispanic respondents had higher odds of being in **Low initial risk with rapid increase**

**Table 2. Odds of cognitive impairment by multimorbidity category in the full group-based trajectory model.**

| | Low risk with late-life increase | Low initial risk with rapid increase | High risk |
|---|---|---|---|
| | OR (95% CI) | OR (95% CI) | OR (95% CI) |
| **Multimorbidity** | | | |
| Somatic multimorbidity | Reference | Reference | Reference |
| No multimorbidity | 0.75 (0.64, 0.88)** | 0.68 (0.62, 0.74)** | 0.74 (0.65, 0.85)** |
| Stroke multimorbidity | 2.06 (1.75, 2.43)** | 2.71 (2.35, 3.13)** | 3.23 (2.16, 4.81)** |
| Depressive multimorbidity | 2.03 (1.71, 2.41)** | 1.89 (1.71, 2.10)** | 1.70 (1.45, 2.00)** |
| Stroke and depressive multimorbidity | 3.92 (2.91, 5.28)** | 4.31 (3.50, 5.31)** | 3.43 (2.07, 5.66)** |

**p<0.01

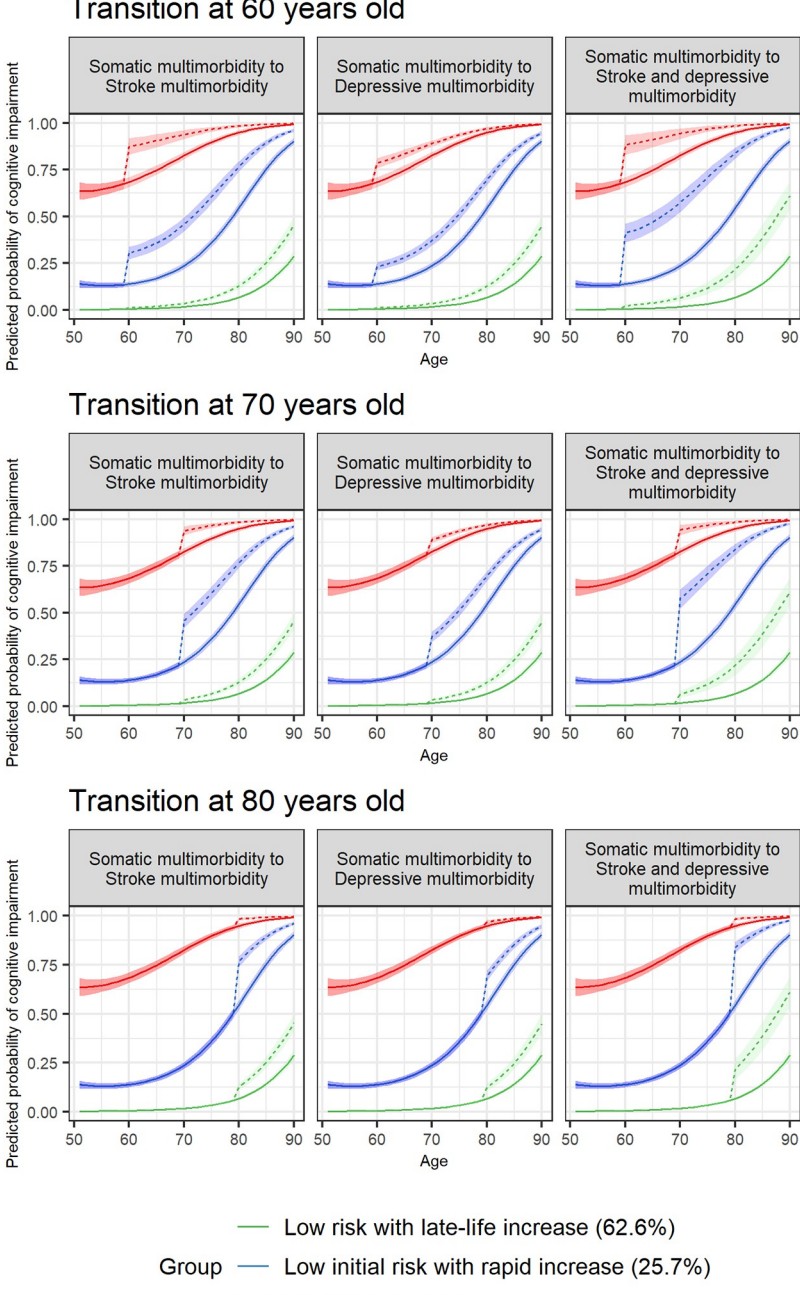

**Fig 2. Predicted probability of cognitive impairment with multimorbidity transition at pre-specified ages in the full group-based trajectory model.** Predicted trajectories in respondents who transitioned from somatic multimorbidity (solid lines) to stroke multimorbidity/depressive multimorbidity/stroke and depressive multimorbidity (dashed lines) at decades of age (60/70/80 years) were shown. Solid lines over the observed age span represent predicted trajectories in respondents with consistent somatic multimorbidity with advancing age. Group membership is assigned for each participant based on maximum posterior probability rule from the full group-based trajectory model (with time-varying multimorbidity, accounting for attrition). Confidence bands represent 95% CIs of predicted probabilities.

**Table 3. Sociodemographic and health-related covariates of trajectory group membership in the multinomial logistic regression model.**

| | Low initial risk with rapid increase | High risk |
|---|---|---|
| **Characteristics** | OR (95% CI) | OR (95% CI) |
| **Race/ethnicity** | | |
| Non-Hispanic White | Reference | Reference |
| Non-Hispanic Black | 2.42(2.19,2.66) ** | 6.17(5.43,7.01) ** |
| Hispanic | 1.59(1.42,1.79) ** | 2.81(2.42,3.26) ** |
| **Sex** | | |
| Male | Reference | Reference |
| Female | 0.81(0.76,0.88) ** | 0.64(0.58,0.72) ** |
| **Education** | | |
| High School Graduate | Reference | Reference |
| <High School | 2.29(2.09,2.51) ** | 6.57(5.84,7.39) ** |
| College | 0.47(0.43,0.52) ** | 0.27(0.22,0.33) ** |
| **Wealth quartiles** | | |
| 4th quartile (highest) | Reference | Reference |
| 3rd quartile | 1.19(1.07,1.32) ** | 1.49(1.20,1.84) ** |
| 2nd quartile | 1.52(1.36,1.68) ** | 1.79(1.46,2.20) ** |
| 1st quartile (lowest) | 1.66(1.48,1.85)** | 3.31(2.71,4.05) ** |
| **Smoking** | | |
| Never smoker | Reference | Reference |
| Past smoker | 0.99(0.92,1.07) | 0.93(0.82,1.05) |
| Current smoker | 1.24(1.13,1.37) ** | 1.13(0.99,1.30) |
| **BMI categories, n (%)** | | |
| Normal weight | Reference | Reference |
| Underweight | 0.91(0.65,1.29) | 0.98(0.59,1.61) |
| Overweight | 0.94(0.86,1.02) | 0.80(0.70,0.91)** |
| Obese | 1.00(0.91,1.10) | 0.78(0.68,0.89)** |

Note: Reference group is Low risk with late-life increase. The model is adjusted for baseline age. Group membership is assigned for each participant based on their maximum posterior probability from the full group-based trajectory model (with time-varying multimorbidity, adjusted for baseline age and accounting for attrition).
**p<0.01

(non-Hispanic Black: OR: 2.42, 95%CI: 2.19, 2.66; Hispanic: OR: 1.59, 95%CI: 1.42, 1.79) and **High risk** (non-Hispanic Black: OR: 6.17, 95%CI: 5.43, 7.01; Hispanic: OR: 2.81, 95%CI: 2.42, 3.26) relative to non-Hispanic White respondents. Similarly, respondents with less than a high school education and lowest wealth quartile were more likely to be in these two groups. Female respondents were less likely to be in these two groups. Current smokers were more likely to be in **Low initial risk with rapid increase**. Overweight and obese categories were associated with lower odds of being in **High risk**.

## 4. Discussion

This longitudinal study of a nationally-representative cohort of adults aged 51 years and older identified three distinct trajectories of the probability of cognitive impairment and quantified the associations of transitions between somatic, mental, and combined mental-somatic multimorbidity with the identified cognitive trajectories. The **Low risk with late-life increase**

trajectory exhibited stable and preserved cognitive function with age, demonstrating a slight increase in the probability of cognitive impairment after age 80. The **Low initial risk with rapid increase** trajectory exhibited a sharp increase in the probability of impaired cognition from age 70 onward. In contrast to the other two groups, the **High risk** trajectory had high probability of cognitive impairment throughout the ages observed. Interestingly, we noted that transitioning to various mental-somatic multimorbidity combinations at different decades of advancing age might be associated with trajectories of cognitive impairment probabilities, with varying degrees of increased risk of cognitive impairment for each trajectory group.

Most notably, our findings indicated that the transition to stroke and depressive multimorbidity was related to the largest upward shift in each identified trajectory compared with transitioning to either stroke or depressive multimorbidity. The observed detrimental joint association of stroke and high depressive symptoms with cognitive decline might be attributed to shared biological abnormalities in the brain resulting from both stroke and depression with a reciprocal relationship [38]: structural changes caused by depression may accelerate the progression of vascular and Alzheimer's neuropathological changes, and conversely, progression of vascular damage in brain may mediate the development of depression. Psychological pathways, such as exposure to stressful life events [39], may also serve as shared mechanisms underlying the impact of depression and stroke on cognition. Furthermore, there are interesting nuances between the three identified trajectories and the consequences of experiencing mental-somatic multimorbidity at different decades of late life. In the trajectories characterized by rapidly increasing cognitive impairment risk (**Low initial risk with rapid increase**) or sustained **High risk** in mid-life, onset of stroke multimorbidity accounts for much of the increase in risk of cognitive impairment. For adults following along the best performing trajectory (**Low risk with late-life increase**), the increased risk of cognitive impairment appears to be equivalent when transitioning to stroke/depressive/stroke and depressive multimorbidity, although stroke and depressive multimorbidity has a greater association with the risk of cognitive impairment in late life.

Understanding the heterogeneity among identified cognitive trajectories may aid in understanding characteristics associated with the course of cognitive decline. The multinomial regression model findings suggesting that racial/ethnic disparities and variation in educational attainment may differentially contribute to cognitive outcomes among older adults were consistent with existing literature [40, 41]. The highest odds of being in the **High risk** trajectory observed for minoritized adults and adults with less than a high school education suggests that persistently worse cognitive function might be partially explained by systemic and structural disparities and inequities imposed by social and environmental factors starting from early life [42, 43]. The **Low initial risk with rapid increase** trajectory was also found to be associated with individuals from Black and Hispanic backgrounds, social disadvantages from having lower wealth, and being less educated, indicating that the acceleration of cognitive change in the trajectory might be associated with a number of factors, such as less resilience to age-related changes in health, and diminished ability to tap into protective socio-economic resources as a result of lower wealth streams and educational inequities earlier in life. Collectively, these findings demonstrate that adults with lower wealth, lower educational background and minoritized groups may be more susceptible to somatic-mental multimorbidity-related cognitive impairment pathways, and add to existing literature by examining this extended association over a substantial period throughout middle and late life. However, we found that adults with obesity and overweight status had a lower likelihood to be a member in the **High risk** trajectory, indicating a potential protective effect of higher BMI on cognitive decline, which was also observed in other published research [44–46]. This "paradox" is possibly attributable to better nutritional status [47] and lower expression and deposition of AD-related

biomarkers such as Aβ [46] in higher BMI groups. Moreover, the physical activity theory supports that people with obesity are more inclined to perform physical activities that can help protect cognitive function in later life [48].

Consistent with our findings, several studies investigating heterogeneity of cognitive trajectories in U.S. population across a variety of data sources [49] identified a number of trajectory groups with similar distinct courses [49–53]. Considering the high prevalence of multimorbidity among adults at increased risk for dementia, it is noteworthy that specific combinations of multimorbidity may be differentially associated with cognitive decline [12]. Chen et.al (2022) evaluated the association of multimorbidity burden and developmental trajectories of later-life dementia [54] and found that higher multimorbidity burden at baseline and rapid growth in the number of chronic conditions was associated with higher risk of dementia. Our study corroborates and adds to these findings by quantifying the role of stroke and depressive symptoms in the context of multimorbidity. Importantly, rather than evaluating multimorbidity and dementia in two separate periods of time, our study identifies distinct cognitive courses by applying the group based trajectory modeling approach to estimate contemporaneous changes in multimorbidity profiles and their associations with cognition from mid to late life.

Our study has several strengths. First, the HRS provides large, rich, and longitudinal survey data that enables us to model progression of cognitive impairment over an extended period, starting in middle age and into late life. Second, the prospective design and modeling approach with time-varying covariates allows us to evaluate changes in mental-somatic multimorbidity combinations and estimate associated probabilities of cognitive impairment. Third, our study adds to the emerging literature by examining the development of clinically meaningful mental-somatic multimorbidity combinations and its associations with risk of cognitive impairment over a substantial period in mid and late life.

A few limitations should also be noted. First, the physician-diagnosed chronic condition data are self-reported, and those reporting a stroke would have been survivors. Additionally, HRS collected depressive symptoms and not clinically diagnosed depression and its various subtypes. Similarly, cognitive function is measured with the validated TICS assessment instead of clinically diagnosed dementia. However, several studies have demonstrated concordance between respondent reports of conditions and ascertained disease diagnosis from other resources [55]. Second, given the long observation period in HRS design, the survival bias should be noted in the analyses. To mitigate the potential bias due to healthy survivorship, we extended the GBTM model to account for nonrandom attrition and conducted sensitivity analysis. Finally, while it is imperative to study multimorbidity in diverse samples of adults, we were limited by the number of racial and ethnic categories assessed in the HRS. Future studies should examine the risk of dementia associated with mental and somatic multimorbidity changes among even broader numbers of underrepresented racial and ethnic groups using data sources that facilitate these analyses.

Our study has important implications. We found that development of mental-somatic multimorbidity combinations with both high depressive symptoms and stroke is highly associated with higher probability of cognitive impairment during critical periods of mid to late adulthood. Our study illustrates the importance of specific and targeted efforts for screening for, preventing and treating stroke and depression, and highlights the potential benefits of more tailored interventions particularly in mid-life, as such efforts may contribute to delaying the cognitive decline and reducing the associated personal, societal and informal care burdens and costs of cognitive impairment.

## Supporting information

**S1 Appendix. Comprehensive analysis documentation.**
(DOCX)

**S2 Appendix. Technical details and reproducible codes for visualization of cognitive impairment trajectories with multimorbidity transition.**
(DOCX)

## Author Contributions

**Conceptualization:** Siting Chen, Corey L. Nagel, Ruotong Liu, Anda Botoseneanu, Heather G. Allore, Jason T. Newsom, Stephen Thielke, Jeffrey Kaye, Ana R. Quiñones.

**Formal analysis:** Siting Chen.

**Funding acquisition:** Ana R. Quiñones.

**Methodology:** Siting Chen, Corey L. Nagel, Heather G. Allore, Jason T. Newsom, Ana R. Quiñones.

**Project administration:** Ana R. Quiñones.

**Supervision:** Corey L. Nagel, Heather G. Allore, Ana R. Quiñones.

**Visualization:** Siting Chen.

**Writing – original draft:** Siting Chen.

**Writing – review & editing:** Corey L. Nagel, Ruotong Liu, Anda Botoseneanu, Heather G. Allore, Jason T. Newsom, Stephen Thielke, Jeffrey Kaye, Ana R. Quiñones.

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
