## [Decision Letter · Decision Letter 0]

11 Mar 2024

PONE-D-23-40847Mental-Somatic Multimorbidity in Trajectories of Cognitive Function for Middle-Aged and Older AdultsPLOS ONE

Dear Dr. Quiñones,

Thank you for submitting your manuscript to PLOS ONE. After careful consideration, we feel that it has merit but does not fully meet PLOS ONE’s publication criteria as it currently stands. Therefore, we invite you to submit a revised version of the manuscript that addresses the points raised during the review process. The manuscript has been assessed by two reviewers. Their comments are appended below. The reviewers have raised some of minor concerns about the manuscript, and in particular they feel that some methodological issues exist that affect the technical soundness of your study, and the conclusions of the paper.

We look forward to receiving your revised manuscript.

Kind regards,

Bruno Pereira Nunes, Ph.D.

Academic Editor

PLOS ONE

Journal Requirements:

Reviewers' comments:

Reviewer's Responses to Questions

**Comments to the Author**

1. Is the manuscript technically sound, and do the data support the conclusions?

Reviewer #1: Yes

Reviewer #2: Partly

2. Has the statistical analysis been performed appropriately and rigorously? 

Reviewer #1: Yes

Reviewer #2: N/A

3. Have the authors made all data underlying the findings in their manuscript fully available?

Reviewer #1: Yes

Reviewer #2: Yes

4. Is the manuscript presented in an intelligible fashion and written in standard English?

Reviewer #1: Yes

Reviewer #2: Yes

5. Review Comments to the Author

Reviewer #1: This is a high-quality manuscript. The study is novel, interesting, and carefully reported. The research question is important and the methods for testing are rigorous. I enjoyed reading this report and unreservedly recommend it be published.

I have suggestions for fairly minor revisions, focused on grammar and presentation.

Section 2.3.3. starts with a description of statistical analysis that be deleted and instead placed in Section 2.4. In its current form, it disrupts the reader's flow.

Typo in 2.4.1.- " Following established guidance [31], We began by fitting a sequence of unconditional GBTMs in order to 1)", we should be in lowercase.

Section 2.4. In its current form is a bit hard to read. You can change this by creating paragraph separations. E.g. in 2.4.2 - "Moreover, changes in multimorbidity..." should be a separate paragraph.

Excellent job at providing detailed descriptions of the methods in the Appendix section.

Section 3.2. Place the group size immediately after the group name for consistency. E.g., "Low risk with late-life increase trajectory (63.7%), represented the largest proportion of the study sample "

Section 3.5. Please re-state in sentence form "respondents with < high school education"

The discussion section is well-framed and contextualised.

Reviewer #2: The study explored associations between mental-somatic multimorbidity combinations with both high depressive symptoms and stroke and probability of cognitive impairment during adulthood. I have some comments and questions about this article:

2.2. Study Population

The authors excluded participants who reported other races like Asians, African Americans and Native Americans.

Question 1- Why exclude these participants who represent 9% of the racial composition of Oregon's population? It’s a standard amongst researchers to include all categories of race in the study’s region

2.3.3 Covariates

Question 2- Why, in this topic, wasn’t the male gender reported, only the female?

Question 3- The topic about including/excluding races was not clear (mutually exclusive categories: non-Hispanic White, non-Hispanic Black, Hispanic). Could you explain it further?

2.4. Statistical analysis

The GBTM is an appropriate model for this type of study, as it relies on data to generate latent subgroups of individuals with different health trajectories over time and, consequently, potentially differential risks of the disease. Although the statistical analysis used to estimate differences between groups has a few limitations:

- Table 1 provides detailed descriptive information on the analytic sample at baseline. The main limitation observed in this table is not presenting the percentage by rows, only by columns. Without this information, it is not possible to infer the distribution of the outcome variables between the independent variables or covariates.

Question 4- Was there a statistically significant difference between groups by sex, race, age, etc?

Question 5- For example: In table 1, among men, what is the prevalence in each of the Three distinct cognitive trajectories?

Question 6- Was there a statistically significant difference between the prevalence of men and women within the groups?

Question 7- Was there a statistically significant difference in the risk of individuals in the group without multimorbidity being included in each stage when compared to individuals with multimorbidity?

Regarding references, I suggest bringing more recent articles. 50% were more than 8 years old since its publication. There is a lot of scientific literature on multimorbidity, depressive symptoms and cognitive impairment.

6. PLOS authors have the option to publish the peer review history of their article (what does this mean?). If published, this will include your full peer review and any attached files.

Reviewer #1: No

Reviewer #2: No

---

## [Author Response · Author response to Decision Letter 0]

22 Apr 2024

Reviewer #1: 

This is a high-quality manuscript. The study is novel, interesting, and carefully reported. The research question is important and the methods for testing are rigorous. I enjoyed reading this report and unreservedly recommend it be published. I have suggestions for fairly minor revisions, focused on grammar and presentation.

We thank the reviewer for their kind remarks.

1. Section 2.3.3. starts with a description of statistical analysis that be deleted and instead placed in Section 2.4. In its current form, it disrupts the reader's flow.

We have made this change according to the reviewer’s suggestion.

2. Typo in 2.4.1. " Following established guidance [31], We began by fitting a sequence of unconditional GBTMs in order to 1)", we should be in lowercase.

We have made this change according to the reviewer’s suggestion.

3. Section 2.4. In its current form is a bit hard to read. You can change this by creating paragraph separations. E.g. in 2.4.2 - "Moreover, changes in multimorbidity..." should be a separate paragraph. Excellent job at providing detailed descriptions of the methods in the Appendix section.

We have made this change according to the reviewer’s suggestion.

4. Section 3.2. Place the group size immediately after the group name for consistency. E.g., "Low risk with late-life increase trajectory (63.7%), represented the largest proportion of the study sample ".

We have made this change according to the reviewer’s suggestion.

5. Section 3.5. Please re-state in sentence form "respondents with < high school education"

We have revised the sentence to, “respondents with less than a high school education”.

 

Reviewer #2

The study explored associations between mental-somatic multimorbidity combinations with both high depressive symptoms and stroke and probability of cognitive impairment during adulthood. I have some comments and questions about this article:

2.2. Study Population

1. The authors excluded participants who reported other races like Asians, African Americans and Native Americans. Why exclude these participants who represent 9% of the racial composition of Oregon's population? It’s a standard amongst researchers to include all categories of race in the study’s region.

We agree with the reviewer’s point regarding the importance of inclusivity and diversity in research. However, we opted not to include “other” race category in our analytic sample because the publicly available race category includes a highly heterogeneous composition of study participants, including American Indian, Alaskan Native, Asian, Native Hawaiian, and Pacific Islander, and any other races, rendering any differences of this category difficult to interpret. As a result, we decided to include mutually exclusive categories (for Non-Hispanic White, Non-Hispanic Black and Hispanic) in our analyses, and provide further details and explanation regarding how racial/ethnic categories were constructed. We also note this as a limitation to our work and call for future work to examine mental-somatic multimorbidity among underrepresented racial and ethnic groups with data that facilitate this line of inquiry: “Finally, while it is imperative to study multimorbidity in diverse samples of adults, we were limited by the number of racial and ethnic categories assessed in the HRS. Future studies should examine the risk of dementia associated with mental and somatic multimorbidity changes among even broader numbers of underrepresented racial and ethnic groups using data sources that facilitate these analyses (pg. 21).” 

2.3.3. Covariates

2. Why, in this topic, wasn’t the male gender reported, only the female?

We have included descriptive information for male sex in Table 1 and made the change in 2.3.3 Covariates and 3.1. Sample characteristics.

3. The topic about including/excluding races was not clear (mutually exclusive categories: non-Hispanic White, non-Hispanic Black, Hispanic). Could you explain it further?

We have provided an additional brief explanation of race/ethnicity category in 2.3.3 Covariates. The race/ethnicity was defined according to the respondent’s answers to the following questions: 1) “Do you consider yourself Hispanic or Latino?” and 2) “Do you consider yourself primarily white or Caucasian, Black or African American, American Indian, or Asian, or something else?” The answer to the first question was prioritized, which means self-identification of Hispanic (or Latino) was given precedence over any other racial categories and the respondent would be categorized as Hispanic (or Latino) if the study participant answered “yes”. If the respondents didn’t consider themselves as Hispanic or Latino (answered “No” for the first question), they were categorized into Non-Hispanic White or Non-Hispanic Black based on their answers to the second question - white or Caucasian, or, Black or African American. We have provided a table in the attached response letter showing how the mutually exclusive racial/ethnic categories were constructed.

2.4. Statistical analysis

The GBTM is an appropriate model for this type of study, as it relies on data to generate latent subgroups of individuals with different health trajectories over time and, consequently, potentially differential risks of the disease. Although the statistical analysis used to estimate differences between groups has a few limitations:

Table 1 provides detailed descriptive information on the analytic sample at baseline. The main limitation observed in this table is not presenting the percentage by rows, only by columns. Without this information, it is not possible to infer the distribution of the outcome variables between the independent variables or covariates.

We have provided an additional table as S1 Table in S1 Appendix presenting the distribution of the trajectory groups by levels of independent variables and covariates. We performed additional statistical tests for comparisons and added the results to the table as well. The table was also provided in the attached response letter.

While we agree with the reviewer that the trajectory group is the main outcome variable in our multinomial regression analysis and it can provide additional information to present the distribution of trajectory groups by levels of covariates (the table shown above), one of our primary aims of presenting the distribution of sociodemographic and health-related covariates (e.g. racial/ethnicity, education, wealth, etc.) by trajectory groups in Table 1 is to show differences in the proportions of racial/ethnic minoritized groups and socioeconomic characteristics by the different trajectory groups. In this way we provide additional descriptive information prior to presenting results from the analyses. Therefore, we opted to keep Table 1 formatted as is in the manuscript but did add one additional column to present the results from statistical tests comparing the groups. We have also added relevant description of statistical tests in 2.4.3 in Statistical Analysis section. We have provided an additional table as S1 Table in S1 Appendix presenting the distribution of the trajectory groups by levels of independent variables and covariates. However, if the Editor/Reviewer prefers that S1 Table be in the manuscript we are willing to change them.

4. Was there a statistically significant difference between groups by sex, race, age, etc.?

 There were statistically significant differences between groups in terms of sex (p<0.01), race (p<0.01) and age (p<0.01). We have performed the statistical tests and presented the results in both Table 1 in the manuscript and Table S1 in S1 Appendix.

5. For example, in table 1, among men, what is the prevalence in each of the three distinct cognitive trajectories?

 Among men, 61.3% were in the Low risk with late-life increase group, 26.1% in the Low initial risk with rapid increase group, and 12.7% in the High risk group. The information is in Table S1 in S1 Appendix.

6. Was there a statistically significant difference between the prevalence of men and women within the groups? 

 We have performed the statistical tests and there was a statistically significant difference in male and female prevalence within the groups (p<0.01).

7. Was there a statistically significant difference in the risk of individuals in the group without multimorbidity being included in each stage when compared to individuals with multimorbidity?

We included a table in the attached response letter showing the distribution of No multimorbidity and Multimorbidity (collapsing all four multimorbidity categories) at baseline. We performed additional statistical tests to compare between No multimorbidity with Multimorbidity at baseline: the proportion of individuals with multimorbidity at baseline is higher in the High risk group (14.1% vs. 9.9%, p<0.01) and in the Low initial risk with rapid increase group (30.0% vs. 22.2%, p<0.01) when compared with individuals without multimorbidity at baseline.

 Additionally, we performed a multinomial regression analysis including baseline multimorbidity (yes or no) as an independent variable while adjusting for other covariates. This additional analysis shows that individuals with multimorbidity at baseline had a higher probability of being included in the Low initial risk with rapid increase group (OR:1.19, 95%CI: 1.11,1.28) and High risk group (OR:1.12, 95%CI: 1.01, 1.25) when compared with individuals with no multimorbidity at baseline. However, we opted not to include a binary variable for multimorbidity at baseline, as our analyses are not cross-sectional; thus, extended beyond this specification by examining the time-varying nature of multimorbidity on the probability of cognitive impairment over time with age as the time scale. As a result, we include multimorbidity in the group-based trajectory model as a changing category which was not static over time for individuals to account for its association with cognitive impairment probability within each trajectory, rather than include a one-time measurement of baseline multimorbidity to predict the group membership.

8. Regarding references, I suggest bringing more recent articles. 50% were more than 8 years old since its publication. There is a lot of scientific literature on multimorbidity, depressive symptoms and cognitive impairment.

 In addition to some foundational references initially included, we added more recent literature in the citations for the manuscript to update the introduction and discussion and bring more updated findings to our presentation of the topic:

1. Zhang XX, Tian Y, Wang ZT, Ma YH, Tan L, Yu JT. The Epidemiology of Alzheimer's Disease Modifiable Risk Factors and Prevention. J Prev Alzheimers Dis. 2021;8(3):313-21

2. Kadambi S, Abdallah M, Loh KP. Multimorbidity, Function, and Cognition in Aging. Clin Geriatr Med. 2020;36(4):569-84.

3. Huang YY, Chen SD, Leng XY, Kuo K, Wang ZT, Cui M, et al. Post-Stroke Cognitive Impairment: Epidemiology, Risk Factors, and Management. J Alzheimers Dis. 2022;86(3):983-99

4. Yang L, Deng YT, Leng Y, Ou YN, Li YZ, Chen SD, et al. Depression, Depression Treatments, and Risk of Incident Dementia: A Prospective Cohort Study of 354,313 Participants. Biol Psychiatry. 2023;93(9):802-9

5. Quinones AR, Nagel CL, Botoseneanu A, Newsom JT, Dorr DA, Kaye J, et al. Multidimensional trajectories of multimorbidity, functional status, cognitive performance, and depressive symptoms among diverse groups of older adults. J Multimorb Comorb. 2022;12:doi: 10.1177/26335565221143012

---

## [Editor Report · Decision Letter 1]

29 Apr 2024

Mental-Somatic Multimorbidity in Trajectories of Cognitive Function for Middle-Aged and Older Adults

PONE-D-23-40847R1

Dear Dr. Quiñones,

We’re pleased to inform you that your manuscript has been judged scientifically suitable for publication and will be formally accepted for publication once it meets all outstanding technical requirements.

Kind regards,

Bruno Pereira Nunes, Ph.D.

Academic Editor

PLOS ONE
---

## [Editor Report · Acceptance letter]

2 May 2024

PONE-D-23-40847R1 

PLOS ONE

Dear Dr. Quiñones, 

I'm pleased to inform you that your manuscript has been deemed suitable for publication in PLOS ONE. Congratulations! Your manuscript is now being handed over to our production team.

Kind regards, 

on behalf of

Dr. Bruno Pereira Nunes 

Academic Editor

PLOS ONE